# Deep Active Latent Surfaces for Medical Geometries

Patrick M. Jensen[*1], Udaranga Wickramasinghe[2], Anders B. Dahl[1], Pascal Fua[†3], and Vedrana A. Dahl[†1]

[1]Technical University of Denmark, Kgs. Lyngby, Denmark
[2]Advanced Interactive Systems, Lausanne, Switzerland
[3]EPFL, Lausanne, Switzerland
patmjen@dtu.dk, udaranga@gmail.com, abda@dtu.dk, pascal.fua@epfl.ch, vand@dtu.dk

## Abstract

Shape priors have long been known to be effective when reconstructing 3D shapes from noisy or incomplete data. When using a deep-learning based shape representation, this often involves learning a latent representation, which can be either in the form of a single global vector or of multiple local ones. The latter allows more flexibility but is prone to overfitting. In this paper, we advocate a hybrid approach representing shapes in terms of 3D meshes with a separate latent vector at each vertex. During training the latent vectors are constrained to have the same value, which avoids overfitting. For inference, the latent vectors are updated independently while imposing spatial regularization constraints. We show that this gives us both flexibility and generalization capabilities, which we demonstrate on several medical image processing tasks.

## 1 Introduction

3D shape reconstruction from noisy or incomplete data often benefits from shape priors. With deep-learning, this usually means searching for an appropriate latent representation under regularization losses. This representation can be a single global latent vector per shape or a grid of latent vectors.

In both cases, balancing the quality of fit against regularization strength is challenging. Too much regularization results in overly smooth shapes. Too little damages robustness and generalization. Our insight is to use multiple latent vectors but impose the regularity constraints on the latent vectors *instead of* the surface. This is effective because, if a latent vector models a sharp feature, requiring that this vector be similar to its neighbors will not detract from that. To this end, we represent surfaces as triangulated meshes and advocate using separate latent vectors at each vertex. We then impose smoothness on the latent vectors by using regularization ideas from early 3D shape modelling [1–3].

More specifically, during training, we use an auto-decoding approach [4] to jointly learn network

*Corresponding Author.
†Equal Supervision

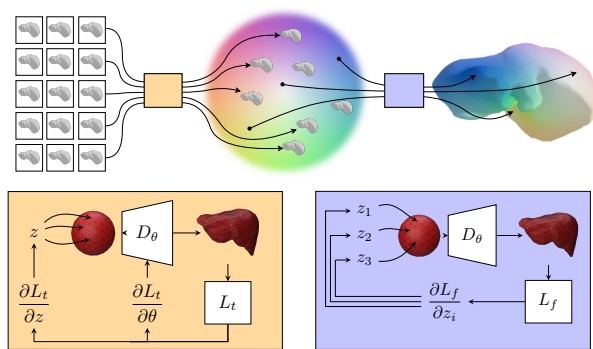

**Figure 1.** *DALS* **overview. (top)** We use an auto-decoding approach to learn a latent space of shapes from a training set. **(bottom left)** Each training shape is associated with a single latent vector **z** that encodes a translation for each vertex of sphere to minimize the distance to the training shape. **(bottom right)** During inference, the latent vectors $\mathbf{z_1}, \mathbf{z_2}, \ldots, \mathbf{z_N}$ at each sphere vertex can change independently to minimize a weighted sum of a data loss function and a regularization term that enforces similarity between neighboring latent vectors.

weights and a *single* latent vector for all vertices of each training sample. At inference time, we then allow the latent vectors to be different at each vertex while enforcing consistency of the vectors via a regularization loss during fitting. Our Deep Active Latent Surfaces (DALS) approach, illustrated in fig. 1, is both easy to train from relatively small datasets and very expressive. By learning a single latent vector per shape, we do not require huge training sets. At model fitting time, we can smoothly blend latent vectors to model complex shapes with sharp features. Finally, using a triangulated mesh instead of a latent vector grid makes our model more compact and simpler to train because it does not have to waste capacity on modeling empty regions.

## 2 Related Work

**Active Surface Models** Active contour models refine contours according to local image properties while remaining smooth. They were first introduced in [1] for interactive delineation and then extended for many different purposes [5]. Active surface mod-

Proceedings of the 6th Northern Lights Deep Learning Conference (NLDL), PMLR 265, 2025.

els [2, 3] replace contours by triangulated meshes to instead model 3D surfaces. They have proved very successful for, e.g., medical [6, 7] and cartographic applications [8], and are still being improved [9–11]. Recently, Deep Neural Networks (DNNs) have been used to evaluate the energy that active contours minimize [12, 13] and to directly predict vertex offsets [14–16]. In [17], active surface models are embedded in special purpose network layers that regularize surface meshes using the same semi-implicit scheme as the original active contours [1].

Balancing data and regularization terms to avoid over-smoothing while being robust to noise remains a challenge. In [15] smoothing is added as a training loss but not during inference. In [17], smoothing is made adaptive to allow sharp edges. Finally, [18] replaces smoothing with preconditioned gradient descent of the external energy. In all these approaches, the regularization tends to flatten sharp geometric features, i.e., over-smooth. In our work, we side step these issues by only smoothing in the latent space.

**Neural Shape Modeling** Deep-learning is now routinely used to model 3D shapes, via models that transform latent vectors into a target shapes parameterized in terms of, e.g., triangulated meshes [19–22], tetrahedral meshes [23, 24], surface patches [25], point clouds [26, 27], voxel grids [28, 29], occupancy functions [30–32], signed and unsigned distance fields [33, 34], or neural splines [35].

Methods may represent a shape with one or several latent vectors. Those that use one obtain the latent vector for a shape with an encoder [30, 36] or by directly optimizing a latent vector [4, 19, 34, 36, 37]. While effective, accurately representing fine details is difficult. Instead, the methods of [32, 33, 38, 39] use a grid of latent vectors and a shared decoder, which greatly increases the model flexibility. However, this wastes latent vectors on representing empty space—increasing memory use and training time. Sparse grids [40, 41] or multiscale tree structures [42, 43] mitigate this but complicate training. Spatial hash encoding [44] implicitly allocates more capacity to surface regions, but is not designed to model multiple surfaces. Our work also relies on multiple latent vectors. However, we store the latent vectors at the vertices of a triangle mesh instead of a spatial grid. This grants us high flexibility but avoids unnecessary latent vectors and a complex training pipeline.

# 3 Deep Active Latent Surfaces

We now describe our Deep Active Latent Surface (*DALS*) approach, illustrated in fig. 1. We represent watertight 3D shapes by triangulated spheres with a latent vector at each vertex. Each, latent vector along with the vertex coordinate, is fed to a decoder $\mathcal{D}_\theta$ that generates an offset vector that translates

the vertex to its final position. After translating all vertices, we have the final shape (see top right of fig. 1). During training, all vertices use the same latent vector. During model fitting, we allow the vectors to be different but impose spatial consistency. This still allows modeling sharp features because these can be predicted by individual latent vectors.

## 3.1 Training Scheme

Formally, let $\mathcal{D}_\Theta$ be a neural network with weights $\Theta$ that takes as input alatent vector $\mathbf{z} \in \mathbb{R}^d$ and a spatial location $\mathbf{x} \in \mathbb{R}^3$ and returns an offset $\mathcal{D}_\Theta(\mathbf{z}, \mathbf{x}) \in \mathbb{R}^3$. Given a triangulated sphere with $V$ vertices and $F$ facets, we denote by $\mathcal{M}_\theta(\mathbf{z})$ the deformed mesh we obtain by translating each vertex $\mathbf{x}_v$ by $\mathcal{D}_\Theta(\mathbf{z}, \mathbf{x}_v)$ for all $v$ between 1 and $V$.

Assume we are given a set of $N$ training shapes $S = \{S_1, \dots, S_N\}$. As in [4], we can simultaneously learn $\Theta$ and a $\mathbf{z}_i$ for each $S_i$ by looking for

$$\Theta^*, \mathbf{z}_1^*, \dots, \mathbf{z}_N^* = \underset{\Theta, \mathbf{z}_1, \dots, \mathbf{z}_N}{\arg\min} \sum_{i=1}^{N} \mathcal{L}_{\mathrm{dat}}(\Theta, \mathbf{z}_i, S_i), \quad (1)$$

$$\mathcal{L}_{\mathrm{dat}}(\Theta, \mathbf{z}, S_i) = L_{\mathrm{cf}}(\mathcal{M}_\theta(\mathbf{z}), S_i) + \\ \lambda_{\mathrm{reg}} L_{\mathrm{reg}}(\mathcal{M}_\theta(\mathbf{z})) + \lambda_{\mathrm{n}} \|\mathbf{z}\|^2, \quad (2)$$

where $L_{\mathrm{cf}}$ is the Chamfer distance [45], $L_{\mathrm{reg}}$ is a shape regularization term, and $\lambda_{\mathrm{reg}}$ and $\lambda_{\mathrm{n}}$ are weighting constants.

In practice, we take $\mathcal{D}_\Theta$ to be an MLP with three hidden layers of size 724, 724, and 362, with ReLU activations for the hidden layers and none for the last layer. Before each ReLU activation we use layer normalization [46]. The input is a concatenation of $\mathbf{x}$ and $\mathbf{z}$. Our initial spherical triangulation is a subdivided icosahedron. We use a pointwise MLP because we want to learn a general mapping from the surface of a sphere conditioned on a latent vector as this allows us to increase the mesh resolution at inference time. To this end, we also randomly rotate the template during training.

The $L_{\mathrm{reg}}$ term in Eq. 2 is intended to encourage the generation of high quality meshes. Experimentally, we found that using the Laplacian did not allow us to generate high quality meshes without losing too many details in the reconstructions. Instead, as in [47], we use

$$L_{\mathrm{reg}}(\mathcal{M}) = 1 - \frac{4\sqrt{3}}{|\mathcal{F}|} \sum_{f \in \mathcal{F}} \frac{A_f}{a_f^2 + b_f^2 + c_f^2}, \quad (3)$$

where $\mathcal{F}$ stands for the mesh facets, $a_f, b_f, c_f$ for the lengths of the three edges of facet $f$, and $A_f$ for its area. $L_{\mathrm{reg}}$ promotes regular triangles without directly penalizing high frequency features [48], as illustrated by fig. 2.

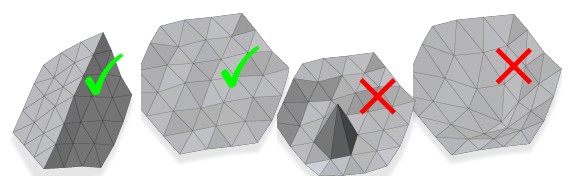

**Figure 2. Behavior of $L_{reg}$.** Minimizing $L_{reg}$ leaves regular meshes (left) unchanged, despite sharp creases. However, for low quality meshes (right), it increases the regularity of the triangles and smoothes isolated outliers.

## 3.2 Fitting Scheme

Let $\mathcal{D} = \mathcal{D}_{\Theta^*}$ be the network we trained in Section 3.1 and let $\mathbf{Z} \in \mathbb{R}^{V \times d}$ whose $V$ rows are latent vectors, one for each vertex of our triangulated sphere. We now denote by $\mathcal{M}(\mathbf{Z})$ the mesh we obtain by shifting each vertex $\mathbf{x}_v$ by $\mathcal{D}(\mathbf{Z}[v])$. In other words, we now assign to each vertex a *different* latent vector and use it to compute the corresponding translation from the initial sphere to where it should be. Our goal is now to find $\mathbf{Z}$ such that the predicted shape $\mathcal{M}(\mathbf{Z})$ solves a given downstream task, e.g., fitting a point cloud. Formally, we look for

$$\mathbf{Z}^* = \underset{\mathbf{Z}}{\arg\min} \, \mathcal{L}_{task}(\mathcal{M}(\mathbf{Z})) + \\ \lambda_{reg} L_{reg}(\mathcal{M}(\mathbf{Z})) + \lambda_{dir} L_{dir}(\mathbf{Z}) \, , \quad (4)$$

where $\mathcal{L}_{task}$ is a task-specific loss function, $L_{reg}$ is the geometric regularization loss of Eq. 3, $L_{dir}$ is a regularization term designed to enforce consistency of the latent vectors across the surface, and $\lambda_{reg}$ and $\lambda_{dir}$ are weighting constants. As an example, to reconstruct a shape from a pointcloud (as in sec. section 4.1), one may take $\mathcal{L}_{task}$ to be the Chamfer distance. However, $\mathcal{L}_{task}$ may be any differentiable loss function. We give additional examples of $\mathcal{L}_{task}$ for our experiments in Section 4.

Inspired by active surfaces [1–3, 17], we use Dirichlet energy [18, 49] to define $L_{dir}$. We write

$$L_{dir}(\mathbf{Z}) = \text{Tr}(\mathbf{Z}^T \mathbf{L}^p \mathbf{Z}) \, , \quad (5)$$

where $\mathbf{L}$ is the uniform Laplacian matrix and $p$ is an integer power. As $\nabla L_{dir}(\mathbf{Z}) = \mathbf{L}^p \mathbf{Z}$, a gradient step corresponds to $p$ iterations of Laplacian smoothing of the latent vectors. We found $p = 2$ to work well.

We study the sensitivity of $\lambda_{reg}$ and $\lambda_{dir}$ in appendix C.5 and found $\lambda_{dir}$ to have the largest effect on fitting results. Weight $\lambda_{dir}$ controls how constrained the fitting is by the prior information modeled by the latent space. When $\lambda_{dir} \to \infty$, all latent vectors will be equal and our approach reverts to a single latent vector approachs. For small values of $\lambda_{dir}$, the model becomes more flexible as the values of the latent vectors can more easily change from vertex to vertex.

In practice, we solve the minimization problem in (4) by initializing each row of $\mathbf{Z}$ to mean of all training latent vectors. Specifically, $\mathbf{Z}_i = (\mathbf{z}_1^* + \mathbf{z}_2^* + \cdots + \mathbf{z}_N^*)/N$ where $\mathbf{z}_i^*$ are the latent vectors found in eq. (1). We then iteratively update the latent vectors using the Adam optimizer.

## 4 Experiments

We demonstrate the benefits of *DALS* on several medical image processing tasks. We train all models to learn a latent representation of livers and spleens using data from the Medical Segmentation Decathlon [50]. To create ground-truth meshes, we resampled the annotated images so that their voxel size is $1 \times 1 \times 1$ mm and used marching cubes [51] to extract isosurfaces. We standardize the surfaces to have zero mean and be contained in the unit sphere.

The datasets contains 111 livers and 41 spleens. We use the first 71 livers for training and hold out the last 40 for evaluation. We also train another model on the first 31 spleens with the last 10 held out for evaluation. We augment the training data using the PointWOLF algorithm [52] to create 100 new shapes for each training shape. PointWOLF applies a smoothly varying non-rigid transformation to mesh vertices yielding diverse and realistic augmentations.

We use 128 dimensional latent vectors and an icosahedron subdivided 3 times for training and 4 times for fitting as a template for the decoder. To learn these vectors and the decoder weights we solve the minimization problem of Eq. 1 with $\lambda_{reg} = 10^{-4}$ and $\lambda_n = 10^{-3}$. We use the ADAM optimizer [53], with learning rate 0.002, momentum terms to $\beta_1 = 0.9, \beta_2 = 0.999$ and train for 24 hours on a single NVIDIA Tesla V100 GPU (ca. 7,500 epochs). If the loss does not improve for 100 epochs we half the learning rate, down to a minimum of $10^{-5}$.

**Baselines** We compare our model against the following baselines: *DeepSDF* [4], *SIREN+DeepSDF* where training and inference are as in *DeepSDF* but with a SIREN [54] based decoder, and the *DUAL-MLP* approach of [39]. These are all auto-decoder based which foregoes the need to train separate encoders for each experiment. We also compare to *DASM*, the active surfaces of [17], along with an improved version that we dub *DASM+R* which adds a re-meshing step during fitting to avoid self-intersections. *DASM* and *DASM+R* do not rely on a learned shape prior and simply promote smoothness.

As the *DUAL-MLP* authors did not release code, we implemented two separate versions of it, one that uses one single latent vector per shape (global) and one that uses several (local). All these methods were trained as recommended in the relevant papers.

Beyond the experiments listed below, we also perform ablation experiments and analyze parameter sensitivity in appendix C.

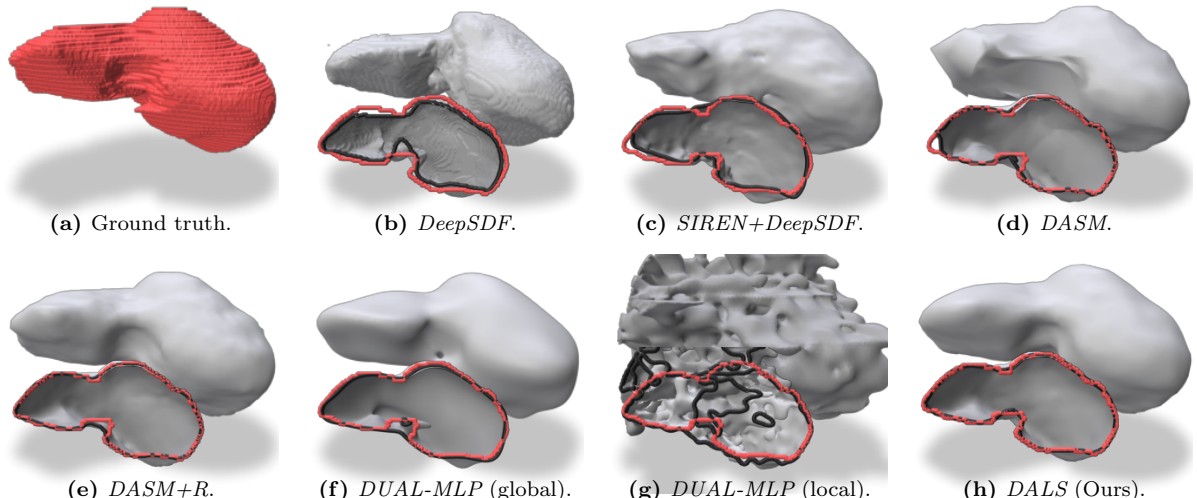

| | | | | |
|:--:|:--:|:--:|:--:|:--:|
| **(a)** Ground truth. | **(b)** *DeepSDF.* | **(c)** *SIREN+DeepSDF.* | **(d)** *DASM.* | |
| **(e)** *DASM+R.* | **(f)** *DUAL-MLP* (global). | **(g)** *DUAL-MLP* (local). | **(h)** *DALS* (Ours). | |

**Figure 3. Reconstruction of a previously unseen liver from 2500 3D points.** For each method, we present the full 3D volume and a version of it cut in the middle. The red outline denotes the ground-truth section and the black one that of the reconstructed organ. Only ours is smooth while still following closely the ground-truth one.

**Table 1. Quantitative results for reconstructing unseen livers from unoriented points.** For each metric we report the mean and standard deviation over the reconstructed shapes. *DALS* concistently produces better reconstructions while still having very good mesh quality. Chamfer distances are multiplied with $10\,000$.

| | Chamfer×10k ↓ | Hausdorff↓ | F@1%↑ | F@2%↑ | Quality↑ | %self. ints.↓ |
|---|:--:|:--:|:--:|:--:|:--:|:--:|
| *DeepSDF* [4] | $40.7 \pm 23.7$ | $0.21 \pm 0.06$ | $31.3 \pm 13.0$ | $63.8 \pm 15.6$ | $\mathbf{0.98 \pm 0.00}$ | $\mathbf{0.00 \pm 0.00}$ |
| *SIREN+DeepSDF* [4, 54] | $36.2 \pm 34.1$ | $0.20 \pm 0.04$ | $51.4 \pm 8.68$ | $78.4 \pm 9.27$ | $\mathbf{0.98 \pm 0.00}$ | $\mathbf{0.00 \pm 0.00}$ |
| *DASM* [17] | $17.0 \pm 10.0$ | $0.23 \pm 0.50$ | $87.7 \pm 3.25$ | $92.9 \pm 2.57$ | $0.74 \pm 0.03$ | $7.40 \pm 4.57$ |
| *DASM+R* [17] | $8.8 \pm 7.53$ | $0.19 \pm 0.07$ | $94.6 \pm 2.57$ | $96.8 \pm 1.97$ | $\mathbf{0.98 \pm 0.00}$ | $\mathbf{0.00 \pm 0.00}$ |
| *DUAL-MLP* (global) [39] | $13.6 \pm 5.67$ | $0.16 \pm 0.05$ | $71.7 \pm 6.39$ | $91.4 \pm 3.48$ | $\mathbf{0.98 \pm 0.00}$ | $\mathbf{0.00 \pm 0.00}$ |
| *DUAL-MLP* (local) [39] | $161.7 \pm 39.6$ | $0.43 \pm 0.04$ | $44.5 \pm 4.05$ | $62.1 \pm 4.08$ | $\mathbf{0.98 \pm 0.00}$ | $\mathbf{0.00 \pm 0.00}$ |
| *DALS* (Ours) | $\mathbf{2.4 \pm 1.04}$ | $\mathbf{0.11 \pm 0.04}$ | $\mathbf{95.4 \pm 2.06}$ | $\mathbf{99.0 \pm 0.76}$ | $\mathbf{0.98 \pm 0.00}$ | $0.20 \pm 0.40$ |

## 4.1 Shape Reconstruction from 3D Point Clouds

**Experimental Setup.** We test the ability of the latent vector models to reconstruct unknown shapes from a given class, here the liver and the spleen, by randomly and uniformly sampling 2,500 points across the test surface and attempting to reconstruct from them by minimizing the loss of Eq. 4. For *DALS*, *DASM*, and *DASM+R* that use a mesh-based representation, we take $\mathcal{L}_{\text{task}}$ to be the Chamfer distance. For the other methods, we take it to be the mean absolute SDF value at the sample points. For all methods, we use the ADAM optimizer to minimize their fitting losses. We set $\lambda_{\text{reg}} = 0.001$ and $\lambda_{\text{dir}} = 0.2$. Finally, for *DASM+R* and *DALS* we post process the results using five iterations of Botsch-Kobbelt remeshing [55].

To evaluate the reconstructions, we use the Chamfer distance, the Hausdorff distance, and the F-score [56, 57] at 1% and 2% of the surface's bounding sphere diameter. We also evaluate the mesh quality of the reconstructions using the quality measure of Eq. 3 and the percentage of self-intersecting faces.

**Results.** We report comparative results in table 1. *DALS* consistently outperforms the other approaches, in part because it can model sharp features more accurately, as can be seen in the qualitative results of fig. 3. Note especially the left side point and the concavity in the lower middle part of the liver. *DALS* also produces excellent mesh quality and keeps the number of intersecting triangles very low although not zero.

We repeated the experiment on much simpler and smoother spleen shapes. The data are again from the Medical Segmentation Decathlon [50] prepared the same way as the liver data. We focused on *DALS* and *DASM+R* since they delivered the best results on the liver. As can be seen in table 3 and fig. 5 *DASM+R* delivers very slightly better metrics but a qualitatively worse reconstruction because it overfits to the staircase artifacts on the ground-truth shape. In contrast, *DALS* yields an organic shape that still fits the data well which can be viewed as a more realistic result. This effect also exists in the liver dataset but did not significantly affect the metrics as the original images were of higher resolution.

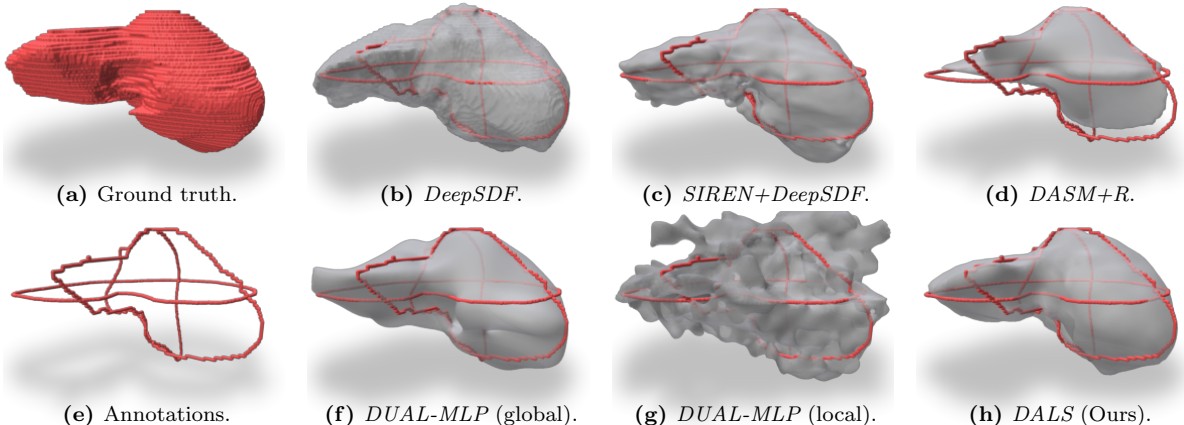

**(a)** Ground truth.     **(b)** *DeepSDF*.     **(c)** *SIREN+DeepSDF*.     **(d)** *DASM+R*.

**(e)** Annotations.     **(f)** *DUAL-MLP* (global).     **(g)** *DUAL-MLP* (local).     **(h)** *DALS* (Ours).

**Figure 4. Reconstruction of a previously unseen liver from outlines in three different planes.** The outlines are shown in **(e)**. Again, our reconstruction is smooth while matching the outlines very accurately.

**Table 2. Quantitative results for reconstructing unseen livers from planar curve annotations.** For each metric (w.r.t. full ground truth) we report the mean and standard deviation over the reconstructed shapes. *DALS* outperforms the baselines while retaining great mesh quality. Chamfer distances are multiplied with 1 000.

| | Chamfer×1k↓ | Hausdorff↓ | F@1%↑ | F@2%↑ | Quality↑ | %self. ints.↓ |
|---|---|---|---|---|---|---|
| *DeepSDF* [4] | $4.36 \pm 2.35$ | $0.22 \pm 0.06$ | $41.5 \pm 12.1$ | $69.5 \pm 12.2$ | $\mathbf{0.98 \pm 0.00}$ | $\mathbf{0.00 \pm 0.00}$ |
| *SIREN+DeepSDF* [4, 54] | $3.82 \pm 2.24$ | $\mathbf{0.21 \pm 0.05}$ | $47.0 \pm 7.67$ | $74.4 \pm 8.25$ | $\mathbf{0.98 \pm 0.00}$ | $\mathbf{0.00 \pm 0.00}$ |
| *DASM+R* [17] | $27.41 \pm 7.91$ | $0.47 \pm 0.09$ | $14.8 \pm 5.99$ | $29.5 \pm 9.99$ | $\mathbf{0.98 \pm 0.00}$ | $0.02 \pm 0.07$ |
| *DUAL-MLP* (global) [39] | $4.05 \pm 1.58$ | $0.23 \pm 0.05$ | $49.9 \pm 6.17$ | $74.4 \pm 6.08$ | $\mathbf{0.98 \pm 0.00}$ | $\mathbf{0.00 \pm 0.00}$ |
| *DUAL-MLP* (local) [39] | $15.55 \pm 4.53$ | $0.39 \pm 0.05$ | $28.3 \pm 2.87$ | $48.3 \pm 4.20$ | $\mathbf{0.98 \pm 0.00}$ | $\mathbf{0.00 \pm 0.00}$ |
| *DALS* (Ours) | $\mathbf{3.27 \pm 1.48}$ | $\mathbf{0.21 \pm 0.05}$ | $\mathbf{52.1 \pm 6.56}$ | $\mathbf{77.2 \pm 6.87}$ | $\mathbf{0.99 \pm 0.00}$ | $\mathbf{0.00 \pm 0.00}$ |

**Table 3. Quantitative results for reconstructing unseen spleens.** Metrics are reported as in Tab. 1.

| | *DASM+R* [17] | *DALS* (Ours) |
|---|---|---|
| Chamfer×10k↓ | $\mathbf{1.6 \pm 0.05}$ | $2.5 \pm 0.89$ |
| Hausdorff↓ | $\mathbf{0.05 \pm 0.02}$ | $0.06 \pm 0.01$ |
| F@1%↑ | $\mathbf{97.8 \pm 1.75}$ | $92.9 \pm 2.73$ |
| F@2%↑ | $\mathbf{100 \pm 0.05}$ | $99.9 \pm 0.11$ |
| Quality↑ | $\mathbf{0.98 \pm 0.00}$ | $\mathbf{0.98 \pm 0.00}$ |
| %self. ints.↓ | $\mathbf{0.00 \pm 0.00}$ | $\mathbf{0.00 \pm 0.00}$ |

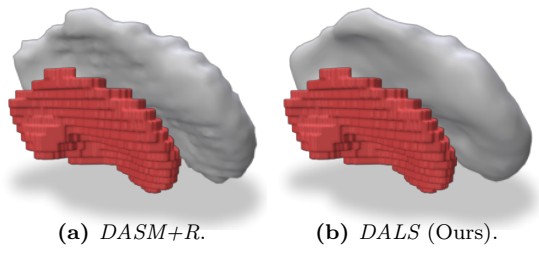

**(a)** *DASM+R*.     **(b)** *DALS* (Ours).

**Figure 5. Reconstruction of previously unseen spleens from 2500 3D points.** The inset shows the ground truth in red.

## 4.2 Shape Reconstruction from Planar Curve Annotations

**Experimental Setup.** When annotating medical images, a common time-saving practice is to only annotate three orthogonal 2D slices instead of the entire 3D image. We test the ability of our model to reconstruct shapes from such weak annotations by fitting to the 2D planar boundary curves extracted from the 2D slice annotations. We randomly sample 5 000 points from the intersection curves between each held out liver and three orthogonal axis-aligned planes. As shown in fig. 4(e), this results in a very sparse point set, making the quality of the embedded shape priors key to obtaining good results.

For *DALS*, *DASM*, and *DASM+R* we take $\mathcal{L}_{\text{task}}$

to be a modified Chamfer distance that relies on distances within the annotation planes (see appendix A for details). For the other methods, we again use the mean absolute SDF value to compute $\mathcal{L}_{\text{task}}$. We again use the ADAM optimizer and Botsch-Kobbelt remeshing as in the previous section. We set $\lambda_{\text{reg}} = 0.01$ and $\lambda_{\text{dir}} = 100$ as we want to rely heavily on the shape prior in this task.

**Results.** We report comparative results in table 2 and qualitative results in fig. 4. To generate these results, we only used annotations in the three orthogonal axis-aligned planes. *DALS* results are consistently better while retaining excellent mesh quality.

**Table 4. Quantitative results for segmentation refinement.** For each metric we report the mean and standard deviation over the 20 reconstructed shapes. Refinement with *DALS* consistently improves the segmentations for all backbones and metrics.

| | Dice↑ Raw | w/ DALS | Hausdorff↓ Raw | w/ DALS | Cham.×10k↓ Raw | w/ DALS |
|---|---|---|---|---|---|---|
| U-Net [58] | $0.81 \pm 0.08$ | **$0.83 \pm 0.08$** | $26.6 \pm 10.2$ | **$20.9 \pm 6.32$** | $44.9 \pm 51.0$ | **$26.0 \pm 22.4$** |
| V-Net [59] | $0.79 \pm 0.16$ | **$0.80 \pm 0.16$** | $28.5 \pm 12.5$ | **$25.4 \pm 9.29$** | $56.4 \pm 74.8$ | **$43.3 \pm 54.8$** |
| nn-U-Net [60] | $0.84 \pm 0.09$ | **$0.85 \pm 0.07$** | $25.2 \pm 11.3$ | **$19.5 \pm 8.08$** | $38.7 \pm 48.4$ | **$22.3 \pm 20.4$** |
| UNETR [61] | $0.74 \pm 0.13$ | **$0.75 \pm 0.17$** | $39.5 \pm 24.2$ | **$26.4 \pm 12.5$** | $164.5 \pm 266$ | **$52.7 \pm 57.0$** |

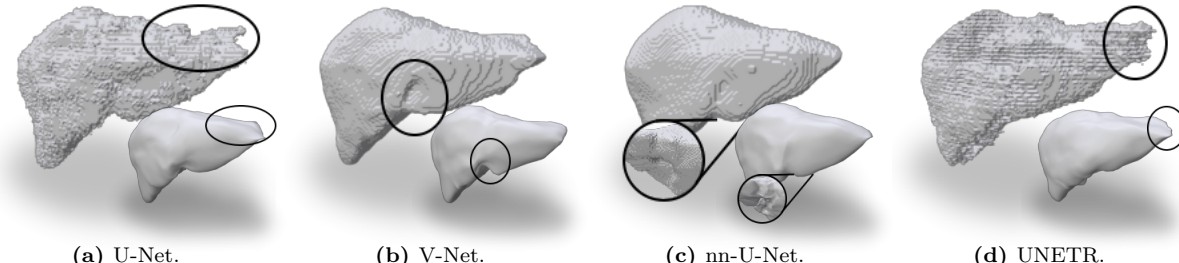

(a) U-Net.  (b) V-Net.  (c) nn-U-Net.  (d) UNETR.

**Figure 6. Comparison of raw (top) and refined (bottom) segmentations.** The black rings highlight examples of *DALS* refinement correcting errors.

This is also the case as one annotates additional planes, which we show in appendix A.1.

### 4.3 3D Image Segmentation with Little Training Data

**Experimental setup** A common medical image analysis task is to segment objects from very few annotations. Here, we use *DALS* to refine a voxel segmentation produced by a CNN backbone network trained on a few 2D slice annotations. As backbones, we use the standard U-Net [58] and V-Net [59], along with the more recent nn-U-Net [60] and UNETR [61].

For this experiment, we use the 40 liver images we held off for testing in the previous experiments. We train the backbones on 20 images and test on the remaining 20. This simulates a realistic scenario with highly limited training data and they have *not* been used to learn the shape priors.

**Results.** As the models are not sufficiently well trained—a common occurrence in medical imaging—the 'raw' segmentations reported in table 4 are suboptimal. To refine them via shape priors, we treat them as noisy data to which we fit a *DALS* model. We initialize the shape at the center and scale predicted by the raw segmentation. We then fit *DALS* to an unsigned distance function computed from the segmentation binary image (details in appendix B). We use $\lambda_{\mathrm{reg}} = 0$, $\lambda_{\mathrm{dir}} = \infty$, and no remeshing to heavily rely on the model's learned prior.

As shown in fig. 6 and table 4, this significantly improves the segmentations w.r.t. both visual appearance and quantitative metrics. Note that *DALS*

removes spurious growths *and* recovers missed concavities, i.e., it does not simply smooth.

## 5 Conclusion

We have shown that *DALS* is an efficient approach for learning a shape model for organic shapes represented as watertight surfaces. We use an autodecoder approach to learn a single latent vector per training shape. However, at model-fitting time, we allow the latent vectors to be different at each vertex

The key ingredient is that we enforce consistency of the latent vectors across the triangulation but *not* on the vertex 3D locations. This enables us to learn the model from a relatively small training set while giving the necessary flexibility to model complex 3D shapes without over-smoothing.

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

# A Fitting *DALS* to Planar Curve Annotations

When annotations are only provided in 2D planes, we only wish to evaluate the reconstruction in these planes. This is similar to how plane annotations are handled for 3D voxel segmentations [58].

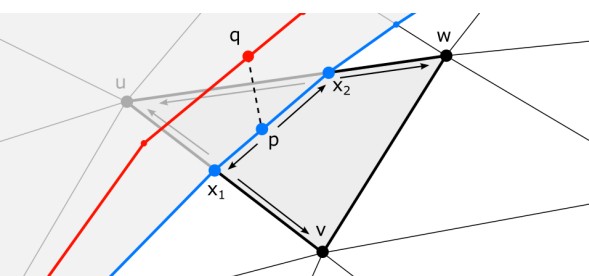

**Figure A.1. Gradient propagation for samples on the intersection between a triangle mesh and a plane.** The blue curve is the intersection of the mesh and the plane, and the red curve is the ground truth boundary curve. The highlighted triangle has vertices $\mathbf{u}$, $\mathbf{v}$, and $\mathbf{w}$ and intersects the plane in the line segment spanned by $\mathbf{x}_1$ and $\mathbf{x}_2$.

Formally, assume we are given a plane $\mathcal{P}$ and a triangle mesh $\mathcal{M}$. To differentially sample points on the intersection between $\mathcal{P}$ and $\mathcal{M}$ we first find the intersection between the plane and each triangle facet. The intersection of a plane and triangle is either empty or a line segment spanned by two points $\mathbf{x}_1$ and $\mathbf{x}_2$. We ignore the degenerate cases where the intersection is the entire triangle or only one of its vertices. We can then sample a point $\mathbf{p}$ on the intersecting line segment as $\mathbf{p} = r\mathbf{x}_1 + (1-r)\mathbf{x}_2$, where $r \in U(0,1)$. Let $(\alpha_1, \beta_1, \gamma_1)$ and $(\alpha_2, \beta_2, \gamma_2)$ be the barycentric coordinates of, respectively, $\mathbf{x}_1$ and $\mathbf{x}_2$. We can then write $\mathbf{p}$ in terms of the triangle vertices $\mathbf{u}$, $\mathbf{v}$, and $\mathbf{w}$ as

$$\mathbf{p} = \begin{bmatrix} \mathbf{u} & \mathbf{v} & \mathbf{w} \end{bmatrix} \begin{bmatrix} \alpha_1 & \alpha_2 \\ \beta_1 & \beta_2 \\ \gamma_1 & \gamma_2 \end{bmatrix} \begin{bmatrix} r \\ 1-r \end{bmatrix}. \qquad (6)$$

As a result, $\mathbf{p}$ is a linear combination of the triangle vertices and $r$ is an independent stochastic term. Therefore, we can propagate a gradient from the point $\mathbf{p}$ to the triangle vertices $\mathbf{u}$, $\mathbf{v}$, and $\mathbf{w}$ [45], see fig. A.1.

Now, let $\mathcal{S}_\mathcal{P}(\mathcal{M})$ denote a set of $M$ points sampled differentially on the intersection of $\mathcal{M}$ and $\mathcal{P}$. Further, let $\mathcal{T}_\mathcal{P}$ be a set of $N$ points sampled uniformly on the planar curve annotations for plane $\mathcal{P}$. In this work we use $M = 5000$ sample points. The loss for plane $\mathcal{P}$ is then

$$L_{\text{cf}}(\mathcal{S}_\mathcal{P}(\mathcal{M}), \mathcal{T}_\mathcal{P}) =$$
$$\frac{1}{M} \sum_{\mathbf{q} \in \mathcal{T}_\mathcal{P}} \min_{\mathbf{p} \in \mathcal{S}_\mathrm{P}(\mathcal{M})} \|\mathbf{p} - \mathbf{q}\|_2^2 + \qquad (7)$$
$$\frac{1}{M} \sum_{\mathbf{p} \in \mathcal{S}_\mathrm{P}(\mathcal{M})} \min_{\mathbf{q} \in \mathcal{T}_\mathcal{P}} \|\mathbf{q} - \mathbf{p}\|_2^2 \,.$$

Note that the above is the Chamfer loss between the plane sample points [45].

Finally, given a collection of planes $\mathcal{P}_1, \mathcal{P}_2, \ldots, \mathcal{P}_P$, the fitting loss, $\mathcal{L}_{\text{task}}$, is

$$\mathcal{L}_{\text{task}}(\mathcal{M}) = \frac{1}{P} \sum_{i=1}^{P} L_{\text{cf}}(\mathcal{S}_{\mathcal{P}_i}(\mathcal{M}), \mathcal{T}_{\mathcal{P}_i}). \qquad (8)$$

## A.1 Additional Results

In fig. A.2, we plot the same quality metrics as in table 2 as a function of the number of annotated planes for one of the livers. *DASM* results are consistently better and improve almost monotonically with the number of planes we provide, which is a very desirable behavior in clinical practice.

# B Fitting *DALS* to Voxel Segmentations

To fit *DALS* to a binary 3D voxel image $\mathbf{B} \in \mathbb{R}^{W \times H \times D}$ we first use the Euclidean distance transform to create a new image $\mathbf{U} \in \mathbb{R}^{W \times H \times D}$ where each voxel contains the unsigned distance to the segmentation boundary. Given a mesh $\mathcal{M}$ with $V$ vertices, the fitting loss is then given by

$$\mathcal{L}_{\text{task}}(\mathcal{M}) = \frac{1}{V} \sum_{v=1}^{V} \mathbf{U}(\mathbf{x}_v), \qquad (9)$$

where $\mathbf{U}(\mathbf{x}_v)$ is trilinear interpolation of $\mathbf{U}$ at vertex position $\mathbf{x}_v$.

To optimize the latent vectors $\mathbf{Z} \in \mathbb{R}^{V \times d}$ we require the gradient of $\mathbf{U}$ at $\mathbf{x}_v$. To get a robust estimate, we use a Sobel operator to pre-compute a gradient image $\mathbf{G} \in \mathbb{R}^{W \times H \times D \times 3}$ which contains the gradient of $\mathbf{U}$ at each voxel position. We then use trilinear interpolation to evaluate the gradient of $\mathbf{U}$ as $\nabla \mathbf{U}(\mathbf{x}_v) = \mathbf{G}(\mathbf{x}_v)$.

We also attempted to fit *DALS* directly to the binary segmentation or the softmax outputs of the CNN backbone. In practice, we found that the distance field gradients made it easier for the model to fit the images.

# C Ablation experiments

To further explore the properties of our model, we perform ablation experiments.

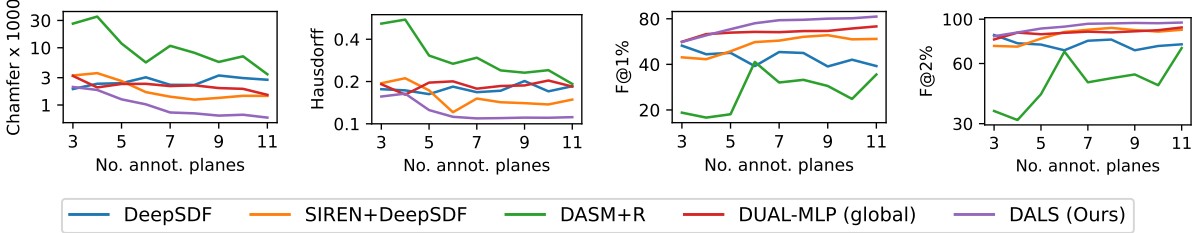

**Figure A.2. Reconstruction metrics for a liver as a function of the number of annotated planes.** Unlike those of other approaches, *DALS* results, shown in purple, consistently improve as more planes are added. However, they tend to saturate after 6 or 7.

## C.1 Local inference and triangle quality loss

We perform an ablation study to investigate how using a single or multiple latent vectors and our triangle quality loss $L_{reg}$ affect performance. Additional ablations can be found in the supplementary materials. To use a single vector, we constrain all latent vectors to be the same during fitting. To remove $L_{reg}$ we set $\lambda_{reg}$ to 0 during training and fitting.

**Table C.1. Quantitative results of the ablation study.** Local inference results in a large boost to reconstruction accuracy and the $L_{reg}$ loss significantly improves triangle quality. Adding remeshing further boosts the triangle quality at some expense to reconstruction accuracy. Chamfer distances are multiplied with 10 000.

| Local | $L_{reg}$ | Remeshing | Chamfer×10k↓ | Quality↑ |
|-------|-----------|-----------|--------------|----------|
|       |           |           | $7.89 \pm 2.75$ | $0.75 \pm 0.03$ |
| ✓     |           |           | $1.89 \pm 0.63$ | $0.72 \pm 0.02$ |
|       | ✓         |           | $5.41 \pm 1.77$ | $0.83 \pm 0.02$ |
| ✓     | ✓         |           | $1.50 \pm 0.49$ | $0.87 \pm 0.01$ |
| ✓     | ✓         | ✓         | $2.41 \pm 1.04$ | $0.98 \pm 0.00$ |

As shown in table C.1, both our local latent vector approach and $L_{reg}$ loss significantly improves reconstruction and mesh quality, even more so when combined. Finally, adding remeshing results in excellent mesh quality at some cost to accuracy, as Botsch-Kobbelt also optimizes vertex positions. In this work, we prioritized mesh quality for our reconstruction results.

## C.2 Local latent vectors

We investigate whether *DALS* actually uses multiple *different* latent vectors during fitting. We use *DALS* to reconstruct a liver from unoriented points as in section 4.1 and illustrate the result in fig. C.1. The livers corresponding to the latent vectors at the four highlighted vertices are noticeably different and the coloring of the reconstruction clearly demonstrates

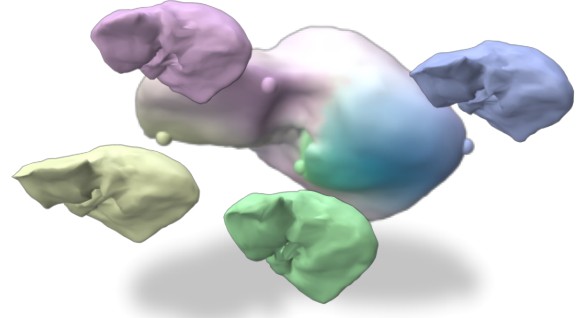

**Figure C.1.** *DALS* **combining multiple shapes.** The large liver is colored according to the latent vector at each vertex. For the highlighted vertices, we show the corresponding decoded livers.

the smooth transition between latent vectors over the surface.

## C.3 Reconstruction time

We compare the reconstruction time for *DALS* against its closest competitors for shape reconstruction: *DUAL-MLP* and *DASM+R*. We use the same setup as in section 4.1 and vary the number of iterations used for fitting. Since each model need a different number of iterations, we only report the time used. Quantitative results are shown in fig. C.2 and *DALS* consistently provides better reconstructions at faster times than the compared baselines.

For the results in the paper, the number of fitting iterations was set high enough to ensure convergence. Specifically, for section 4.1, *DASM+R* used 2.9 seconds, *DUAL-MLP* used 1.3 seconds, and *DALS* used 2.4 seconds.

## C.4 Template mesh resolution

We investigate the effect of the template mesh resolution. Again, we use the same setup as in section 4.1 and vary the resolution of the template. Specifically, we use a subdivided icosahedron subdivided $n = 2, \ldots, 6$ times. Note that we use the same model for all $n$, which was trained using an icosahedron

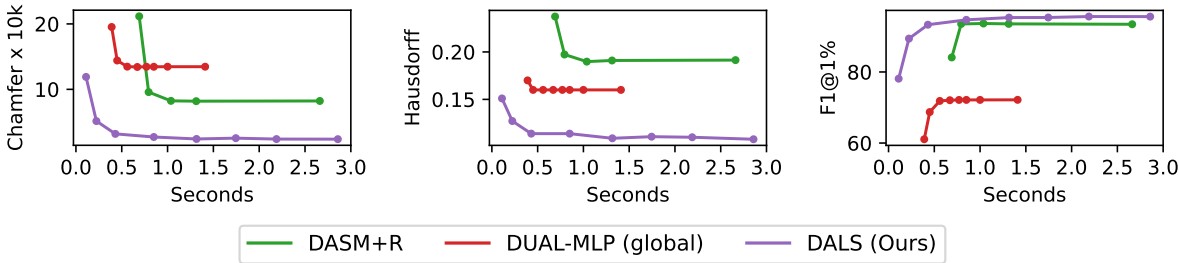

**Figure C.2. Metrics over time for reconstructing unseen livers from 2500 unoriented points**. Each point shows the mean metric over all shapes at the given time. *DALS*, shown in purple, is consistently better than the compared methods.

subdivided 3 times — it is only the template used for fitting that varies. Quantitative results are shown in fig. C.3. Increasing the mesh resolution improves the reconstruction metrics at the cost of higher fitting time and worse mesh quality. We used $n = 4$ subdivision for the experiments in this paper, as we find it provides the best trade-off between reconstruction quality, time, and mesh quality.

## C.5 Sensitivity of fitting regularization

Finally, we explore the sensitivity of the regularization parameters $\lambda_{\text{reg}}$ and $\lambda_{\text{dir}}$ from eq. (4) w.r.t. the quality of the shape reconstruction. We use the same setup as in section 4.1 and vary each regularization parameter while keeping the other constant (set to the value used in the paper). Results are shown in fig. C.4. *DALS* outperforms the second best method for each metric for a wide choice of parameter values, indicating that our *DALS* is not overly sensitive to the choice of regulularization strength.

Regarding $L_{\text{dir}}$, the reconstruction metrics follow a convex shape while the mesh quality (more or less) monotonically improve as the regularization strength is increased—although only slightly. I.e., if the strength is too small, the fitting suffers as the latent vectors have too much freedom and the fitting is less stable. If the strength is too large, the latent vectors have too little freedom to fit the target, but all latent vectors will have more similar values. As the network is trained to generate high quality meshes in the single latent vector case, this then translates to slight higher mesh quality.

Regarding $L_{\text{reg}}$, it has a limited but measurable effect on the metrics for smaller values. As it becomes to large, however, it seems to become more unstable and significantly worsens the reconstruction quality as well as causing more self intersections. The mesh quality, on the other hand, goes up since we are putting more weight on it.

Based on this, our recommendation is to leave $L_{\text{reg}}$ at 0.001 or similar and focus hyperparameter tuning on $L_{\text{dir}}$.

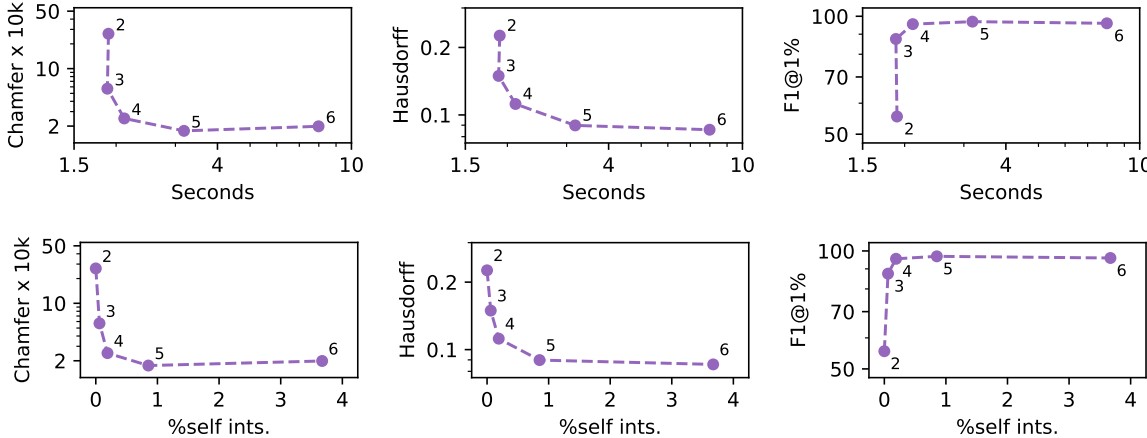

**Figure C.3. Metrics for reconstructing unseen livers for varying mesh resolution.** Reconstructions are based on 2500 unoriented points. Each point shows the mean metric over all shapes using an icosahedron subdivided $n$ times as a template, where $n$ is indicted next to each point. The top row shows metrics over time and the bottom row shows metrics over self intersections.

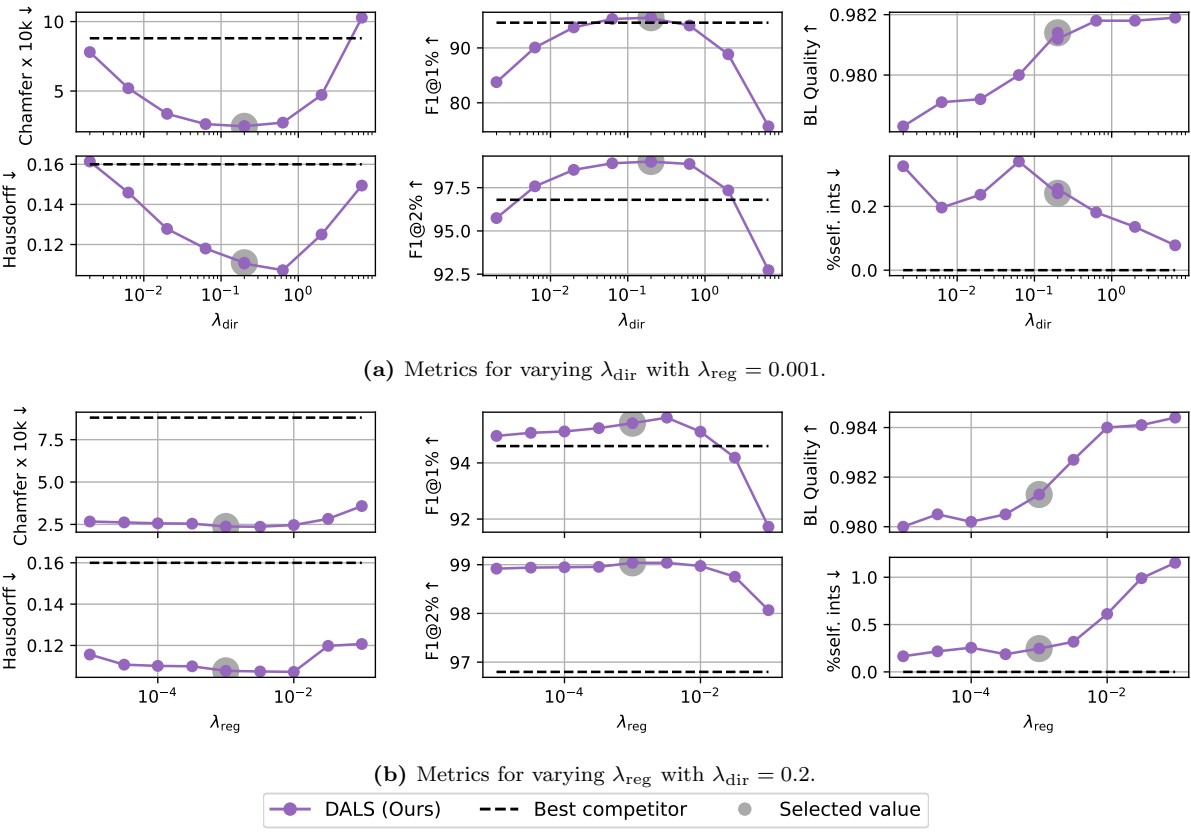

(a) Metrics for varying $\lambda_{\mathrm{dir}}$ with $\lambda_{\mathrm{reg}} = 0.001$.

(b) Metrics for varying $\lambda_{\mathrm{reg}}$ with $\lambda_{\mathrm{dir}} = 0.2$.

— DALS (Ours)  - - - Best competitor  ● Selected value

**Figure C.4. Metrics for reconstructing unseen livers for varying regularization strengths during fitting.** Reconstructions are based on 2500 points. Each point on the purple curve shows the mean metric over all shapes for the given value of $\lambda_{\mathrm{dir}}$ or $\lambda_{\mathrm{reg}}$. The dashed black line shows the value achieved by the second best method for that metric, cf. table 1. The gray circle highlights the value used for the results in section 4.1.

