# OpenReview forum: "Deep Active Latent Surfaces for Medical Geometries"
_NLDL.org/2025/Conference — NLDL 2025 Oral_

### Official Review · Reviewer_byPa · 2024-09-21
**Review "Deep Active Latent Surfaces for Medical Geometries"**

**Confidence:** 4

**Summary:**

The paper proposes a novel auto-decoder method to represent 3D shapes by training a neural net that receives a latent vector and a point on a standard 3D sphere as input and predicts the offset from the 3D sphere vertex location to the actual vertex location for the shape to be modeled. The neural net is a simple 3-layer MLP.

The latent vector is supposed to represent the entire 3D shape and is, therefore, forced to be the same across vertices during training.

At inference time, the optimization is to find latent vectors, separate for each vertex, such that the resulting predicted shape conforms to some loss, such as the agreement with 3D point clouds or human-generated segmentations.

**Strengths:**

The strengths of the paper are:
- With medical geometries, a highly relevant practical application case is addressed.
- The paper competently utilized an entire host of mathematical and practical tools from (neural) geometry processing to generate a novel approach.
- The novel approach is conceptually very clear, elegant, and efficient to an inspiring degree. The paper is a great example how even relatively simple neural network architectures (in this case an MLP) can suffice if the problem is represented in a clever way - and the problem representation in this case strikes me as exceedingly clever.
- The empirical results, both numerically and in terms of pictures shown, are highly promising.

**Weaknesses:**

If the paper has any weakness, then that it contains so much content that it is, sometimes, hard to follow for non-experts in the specific subfield the paper adresses. As such, I would rather like to pose some questions which were not entirely clear to me when reading:
* in line 125, how is x represented? Is it in 3D coordinate space or in angular space?
* is the output of D (i.e. the offset mentioned in line 125) just a scalar, regulating how far the corresponding vertex is from the center of the sphere or does it need to be a 3D vector?
* How were the layer sizes in line 140 found?
* The Fitting Scheme section (3.2) could, I think, profit from an introductory sentence introducing the problem: namely finding a matrix of latent variables Z, such that the predicted shape via D solves some downstream task, such as fitting a point cloud.
* In the Fitting Scheme in 3.2, how is Z initialized?

**Final Rebuttal Confidence:**

4

**Final Rebuttal Justification:**

The reviewers addressed my questions fully and provided additional insight. I stand by the evaluation.

**Justification:**

Overall, the paper strikes me as widely applicable with high-stakes practical application domains, conceptually clever (even beautiful), efficient, and empirically effective. As such, I enthusiastically vote for acceptance. If I could propose a best paper award, I would.

---

> ### Author Rebuttal · Authors · 2024-10-24
>
> Thank you for your kind review and your suggestions for improvement. Below follow our answers and responses to your questions and comments. We have also uploaded an updated version of the paper where our listed changed have been incorporated. Note that all references to lines in our response refers to the original paper.
>
>  * __"In line 125, how is x represented?"__
>
>    It is a 3D coordinate (not angular) and the offset predicted by D is a 3D vector giving the spatial offset. Given that we start from a spherical template, the offset must be a full 3D vector, otherwise we would be limited to represent star convex shapes.
>
>
>    To make this clearer, we have updated the paragraph at L123 to "Formally, let $\mathcal{D}\_\Theta$ be a neural network with weights $\Theta$ that takes as input alatent vector $\mathbf{z}\in\mathbb{R}^d$ and a spatial location $\mathbf{x}\in\mathbb{R}^3$ and returns an offset $\mathcal{D}_{\Theta}(\mathbf{z},\mathbf{x})\in\mathbb{R}^3$"
>
>  * __"How were the layer sizes in line 140 found?"__
>
>    Random search. Early in the project we experimented with different network sizes and these sizes performed well so we stuck with them. However, we do not claim these to be optimal - better sizes may exist.
>
>  * __"The Fitting Scheme section (3.2) could, I think, profit from an introductory sentence [...]"__
>
>    Thank you for this suggestion. We agree and have changed L169 to "Our goal is now to find $\mathbf{Z}$ such that the predicted shape $\mathcal{M}(\mathbf{Z})$ solves a given downstream task, e.g., fitting a point cloud. Formally, we look for"
>
>
>  * __"In the Fitting Scheme in 3.2, how is Z initialized?"__
>
>    We initialize Z such that each row is set to the mean of the training latent vectors. This way, fitting starts from the mean shape and then adjusts to match the given task loss. We have added an additional paragraph at L190 specifying this.

---

### Official Review · Reviewer_8zCr · 2024-10-09
**Powerful and solid contributions to 3D reconstruction**

**Confidence:** 3

**Summary:**

The 3D reconstruction problem is one of the important application cases in computer vision and medical bio-information processing. This paper makes new contributions to the field of 3D reconstruction, while also following the latest trends in the field. One important strategy for conventional 3D reconstruction is the active contour/surface model. However, since these methods directly model the observed 3D shape, it is not easy to deal with noise and regularization during training. Therefore, the proposed method first proposes a new method that applies regularization to the latent variables of the auto-decoder model. Then, in 3D reconstruction, it shows a method for expressing more flexible 3D shapes by utilizing multiple vectors (i.e., a mechanism that can explicitly combine multiple clues that are effective for multiple observed 3D shapes) while satisfying spatial consistency from the auto-decoder obtained in the learning process. As a result, learning can be done efficiently with a small number of samples, and it is well suited to real-world problems in medical bioinformatics processing, such as the 3D shape of the liver used in experiments.

**Strengths:**

- The two major contributions of this paper to the problem of 3D reconstruction are very clear.

1. First, regularization in 3D shape learning is applied to the latent variables expressed by the auto-decoder, rather than to the conventional “observed” shape. This avoids excessive smoothing.

2. Second, instead of using a single latent variable, multiple latent variables are used in the 3D reconstruction. This idea alone has been used in the past. However, when combined with the first contribution, it has the effect of reducing the number of training data in the learning process.

- It is explained that the 3D reconstruction using multiple latent vectors in the proposed method is a generalization of the conventional method using a single vector. Specifically, as explained from line 184, by controlling the weight of a specific term of the objective function, the proposed method can be reduced to the conventional method in special cases. This fact gives us an intuitive insight into why the proposed method works well.

- The survey of previous research is comprehensive, and the issues that the proposed method aims to solve and the motivation behind them are explained very clearly.

- The reproducibility of the proposed method is very high. The hyperparameter settings and how to use third-party tools are explained very clearly, along with their motivations.

- Section 4.2 deals with particularly tough problems specific to biomedical real-world problems (which are somewhat uncommon in broad computer vision tasks), and I can imagine that it will strongly attract the interest of experts in bioinformatics.

**Weaknesses:**

I have not been able to find any fatal flaws in this paper. Thank you for sharing your great contribution. The following are some of the simple questions I had while reading the paper. If there are any items that are important for a deeper understanding of the significance of the proposed method, I would be very grateful if you could let me know.

- I certainly think that the method of expanding the training data 100 times using PointWolf [ICCV2021] is appropriate. On the other hand, is such data augmentation really necessary even for relatively static organs like the liver? For example, for organs like the cardiac heart, which constantly undergo major changes in shape, I have a hunch that such data augmentation would work effectively. However, in the case of the liver, I couldn't easily imagine whether data augmentation would work effectively. I wonder if this was an effective process that was tested experimentally. If the authors had investigated this area in their preliminary study, it would be very useful for readers if they could report it briefly in the supplementary materials, etc. (This is not a critical point, so additional experiments, etc. are not necessary.)

- I found it somewhat difficult to follow the explanation of $L_{reg}$ (this seems to be a classic and well-known method, so it may be because of my lack of prior knowledge). As a result, I was only able to understand it after referring to the original paper [Bhatia&Lawrence, 1990]. However, I also understand that it is difficult to explain it further due to the page limit. If the authors could explain this point using simple illustrations in supplementary materials, etc., it may be possible for people like me with a lack of prior knowledge to follow it immediately.

**Final Rebuttal Confidence:**

3

**Final Rebuttal Justification:**

My slight concerns have all been addressed in the authors' response. In particular, the authors have provided supplementary material to explain the technical details so that this paper can be a self-contained explanation.

**Justification:**

I think this paper is of high quality in terms of clarity of contribution, reproducibility of methods, accuracy of reporting, and clarity of explanation. As it deals with the topic of 3D reconstruction, which attracts a wide range of interest, I imagine that there will be many readers who are interested in it. In addition, the experimental setup used focuses specifically on biomedical 3D reconstruction, so it will also appeal strongly to experts. Therefore, this paper is considered to have value in that it can be shared with the community.

---

> ### Author Rebuttal · Authors · 2024-10-24
>
> Thank you for your review and your suggestions for improvement. Below follow our answers and responses to your questions and comments. We have also uploaded an updated version of the paper where our listed changed have been incorporated. Note that all references to lines in our response refers to the original paper.
>
>  * __"[...] expanding the training data 100 times using PointWolf [...]"__
>
>    This is a great question. We did not investigate the effect of less or additional data augmentation to an extent where we can report on it.
>    It is true that our method may make due with less training samples due the small network size it employs and the static shapes. However, since the compared baselines use deeper networks they may react differently. To keep things simple and comparable, we opted to keep the data the same for all methods.
>
>  * __"I found it somewhat difficult to follow the explanation of L_reg [...]"__
>
>    Thank you for this comment. We have added an illustration to the paper after L159 that illustrates the behavior of this loss. We hope that this helps bring some intuition and makes the explanation easier to understand.

---

### Official Review · Reviewer_gdUj · 2024-10-11

**Confidence:** 4

**Summary:**

This paper proposes DALS - a method for shape reconstruction by learning vertex displacements. The method is based two steps: first, learn an autoencoder for geometric shapes. Second, at inference time, optimize over the latent vectors at each vertex to generate an updated displacement of the vertices, focusing on medical shapes.

**Strengths:**

The paper seems novel to me. Although some of the components are repeated from existing methods, the authors discuss the differences and show improved performance.

The paper is nicely written and easy to follow.

Many visualization are provided.

**Weaknesses:**

The method seems to only work for 0 genus shapes that start from a sphere. This is limited because it cannot represent all shapes. It would be interesting to know how this method can be extended to more shapes.

Missing discussion of similar work in the literature: "Generating 3D faces using Convolutional Mesh Autoencoders".

Missing comparison with classical results. While it is contemporary to use neural networks for reconstructing shapes, there are many classical techniques. Specifically for methods that could benefit medical shapes, there are level-set approaches that were shown to provide good performance, and it would be interesting to know how this method compares with them. Some examples are  "Multimodal 3D Shape Reconstruction under Calibration Uncertainty Using Parametric Level Set Methods‏" and "Parametric Level-sets Enhanced To Improve Reconstruction (PaLEnTIR)"


Questions:

What if you want to support a varying number of vertices? Can the method support it?

How do you ensure the mesh remains valid after translating its vertices?

**Justification:**

The paper is interesting and offers a contribution in medical shape reconstruction. It is well written and has good contribution, so I think it should be accepted.

---

> ### Author Rebuttal · Authors · 2024-10-24
>
> Thank you for your review and your many suggestions for improvement. Below follow our answers and responses to your questions and comments. We have also uploaded an updated version of the paper where our listed changed have been incorporated. Note that all references to lines in our response refers to the original paper.
>
>  * __"The method seems to only work for 0 genus shapes that start from a sphere."__
>
>    This is indeed true, our method currently requires one to choose a genus (in the form of a template) up front. We believe, however, that there are sufficient cases in medical image processing where one is modeling shapes of known topology that our model is still a useful contribution as is.
>
>    To make the topology adaptable would be a major addition which we relegate to future work. We see two potential ways this could be achieved:
>     1. Change the triangle mesh during fitting by, e.g., predicting triangles that can be deleted as in Pan et al., Deep Mesh Reconstruction from Single RGB Images via Topology Modiﬁcation Networks, ICCV, 2019.
>     2. Get rid of the mesh and only keep the points and their normals. After deforming the normed pointcloud, one could then extract a surface mesh using, e.g., Poisson surface reconstruction. This approach is inspired by the method in Peng et al., Shape As Points: A Differentiable Poisson Solver, NeurIPS, 2021.
>
>  * __"Missing discussion of similar work in the literature: "Generating 3D faces using Convolutional Mesh Autoencoders"."__
>
>    Thank you for this reference. We now cite in our related works section in our discussion of neural shape modelling.
>
>  * __"Missing comparison with classical results."__
>
>    We agree that comparing to methods which are not based on deep learning is very beneficial. We already compare to DASM and DASM+R, which do not make use of any learned priors (L229-230). DASM is essentially a modern implementation of ""classical"" active surfaces where the +R version adds remeshing after fitting.
>
>    We also agree that comparing to modern level-set based methods would be interesting, but did not mange to complete this within the time frame of the rebuttal.
>
>  * __"What if you want to support a varying number of vertices?"__
>
>    Yes! Since the offset function D only recieves points (L123-125) but no information about the mesh, one can easily vary the number of vertices (also during training). Indeed, we already make use of this by training with a 3-level subdivided icosahedron and performing inference with a 4-level subdivided icosahedron (L208-210). This was actually our main motivation for using a 'point to offset' function instead of, e.g., a mesh convolutional network. We also did preliminary experiments with adaptively subdividing the template (adding more points to some regions to support more details). However, we found the benefits too minor compared to the added complexity so did not explore this further.
>
>  * __"How do you ensure the mesh remains valid after translating its vertices?"__
>
>    Our method currently has no explicit way of ensuring this. Instead, we rely on the regularisation loss during training and fitting to encourage well formed meshes. We recognize that this is a potential weakness compared to, e.g., level-set approaches.
>
>    However, experimentally, our method tends to produce well formed meshes (c.f., Tab. 1 and Tab. 2) Further, as we show in Table D.1, the regularisation loss significatly improves triangle quality. We also add a Botsch-Kobbelt remeshing step after fitting (L250-251, L287-288) which further improves the results (Tab. D.1)

---

### Official Review · Reviewer_rExU · 2024-10-13
**Interesting approach, method details are not clear enough, evaluated only on one dataset**

**Confidence:** 3

**Summary:**

This paper introduces an active latent shape representation model for shape reconstruction by deforming a triangulated sphere to match the target shape. The proposed approach consists of two stages: training and fitting. While the methodology is intriguing, certain aspects lack clarity and require further elaboration.

**Strengths:**

1. The proposed method is evaluated across various shape reconstruction scenarios, such as 3D point clouds and planar curve annotations.

2. The model consistently performs better than other baselines, particularly in the liver dataset.

**Weaknesses:**

1. Key details of the proposed method are insufficiently explained. For instance, in lines 153-154, the choice of regularization is attributed to the Laplacian regularization being "too smooth," but this might be influenced by the regularization term's weighting. The reasoning behind this choice remains unclear. Additionally, it is ambiguous whether the fitting process is performed in a single step or involves multiple iterations to optimize Eq. 4.

2. The weighting terms in Eq. 4 appear to significantly influence the final results, particularly in terms of smoothness, yet the sensitivity of these terms is not adequately explored. The approach may be sensitive to changes in such hyperparameters. Given the noticeable differences in hyperparameter values between Sections 4.1 and 4.2, the authors should provide a more detailed discussion of how these hyperparameters affect different tasks.

3. The organization of the paper could be improved. For example, the task-specific losses discussed in Section 3.2 should be consolidated into a dedicated section rather than being spread across the experimental sections.

4. It is unclear how the proposed method distinguishes itself from existing literature that uses multiple latent vectors for shape representation. Including an introductory section explaining how shape representation pipelines are typically structured could provide helpful context. Additionally, experimental details (e.g., GPU setup, learning rate) could be moved to an appendix to save some space.

5. To enhance reproducibility, the authors should demonstrate their method on at least one additional dataset, as the current evaluations are all based on a single dataset.

**Final Rebuttal Confidence:**

4

**Final Rebuttal Justification:**

Thank the authors for the rebuttal, which clarified a couple of things in the paper. I am now leaning toward accepting the paper.

**Justification:**

The proposed approach is conceptually interesting, but the current presentation lacks clarity in articulating both the technical contributions and the evaluation methodology.

---

> ### Author Rebuttal · Authors · 2024-10-24
>
> Thank you for your review and your many suggestions for improvement. Below follow our answers and responses to your questions and comments. We have also uploaded an updated version of the paper where our listed changed have been incorporated. Note that all references to lines in our response refers to the original paper.
>
>  * __"Key details of the proposed method are insufficiently explained."__
>
>    Regarding Laplacian regularization, we are referring to the trade-off we also mention at L26-29: finding a balance between mesh quality and reconstruction quality is difficult. When using Laplacian regularization, to achieve the mesh quality we desired, we found we had to place too much weight on the regularization term, which then resulted in meshes that would not capture fine details. This is what we mean by "too smooth". Instead of Laplacian regularization, we use a loss that penalizes triangles that are not regular, but it does not directly punish sharp creases or points. Thus, it encourages good mesh quality without penalizing finer details.
>
>    To make this clearer, we updated L152-154 to: "Experimentally, we found that using the Laplacian did not allow us to generate high quality meshes without losing too many details in the reconstructions." We have also added a figure showing what features the $\mathcal{L}_\text{reg}$ loss will remove as reviewer 8zCr requested.
>
>    Regarding the fitting process, we perform the minimization iteratively using the Adam optimizer (c.f., L247-248 and L287). However, as also indicated by reviewer byPA, this is not clear enough from the current text. We have added an additional paragraph at L190 specifying this.
>
>    Finally, the method one chooses to minimize the fitting loss in (4) is not crucial. During our preliminary experiments we also tried more sofisticated optimizers, but found the added complexity was not worth it. While they could minimize the loss in fewer iterations, the time per iteration was much longer so the overall fitting time was not significantly improved.
>
>  * __"The weighting terms in Eq. 4 appear to significantly influence the final results [...]"__
>
>    Thank you this comment. We agree and have added additional an ablation experiments to the supplementary materials (sec. C.5) exploring how these hyperparameters affect the results for the point cloud fitting task (from Sec 4.1). We now mention these on L235 as: "Beyond the experiments listed below, we also perform ablation experiments and analyze parameter sensitivity in appendix C."
>
>    We note that the parameters we chose for the paper achieve a nice trade-off between reconstruction performance and mesh quality. Furthermore, performance gradually degrades as we move away from the optimal values, but our method remains the best for a wide range of values. This indicates that the values of these parameters are important for the perfomance, but we are not overly sensitive to them. We thank you for suggesting this experiment. Finally, we found that $\mathcal{L}\_\text{dir}$ has a significantly higher effect on the reconstruction quality than $\mathcal{L}\_\text{reg}$ and we therefore recommend that users focus any hyperparameter tuning here. We now mention this on L187 as: "We study the sensitivity of $\lambda_\text{reg}$ and $\lambda_\text{dir}$ in appendix C.5 and found $\lambda_\text{dir}$ to have the largest effect on fitting results." We provide additional comments in the appendix (sec. C.5)
>
>  * __"[...] the task-specific losses discussed in Section 3.2 should be consolidated [...]"__
>
>    We agree that leaving the task specific loss L_task in eq. (4) completely open is too vague and likely confusing for the reader. We have changed the text to give the example of using Chamfer distance as a loss when recostructing from point clouds. Specifically, we changed the text on L175-176 to: "As an example, to reconstruct a shape from a pointcloud (as in sec. 4.1), one may take $\mathcal{L}\_\text{task}$ to be the Chamfer distance. However, $\mathcal{L}\_\text{task}$ may be any differentiable loss function. We give additional examples of $\mathcal{L}\_{\text{task}}$ for our experiments in Section 4."
>
>    However, we would prefer not to consolidate the explanation of all losses under the method section. A major point of our method is that, once trained, the model can be used for a wide variety of downstream tasks. We give examples of tasks in our experiments, but this is not an exhaustive list. Therefore, the specific definition of the task-specific loss is not part of our method per se, but more closely tied to the downstream task at hand. This is our reason for defining the task-specific losses as part of the experimental setup and not under the method section.
>
>  * __"It is unclear how the proposed method distinguishes itself from existing literature that uses multiple latent vectors [...]"__
>
>    We discuss existing approaches to shape modelling with multiple latent vectors in our related works section on L97-107, but agree that we could highlight the differences to our method more. The most significant difference is that, instead of storing our latent vectors in a spatial grid (or some other complex spatial datastructure), we use a triangle mesh and store the latent vectors at the mesh vertices.
>
>    To clarify this, we have changed the text on L106-107 to: "Our work also relies on multiple latent vectors. However, we store the latent vectors at the vertices of a triangle mesh instead of in a spatial grid. This grants us high flexibility but avoids unnecessary latent vectors and a complex training pipeline."
>
>  * __"To enhance reproducibility, the authors should demonstrate their method on at least one additional dataset [...]"__
>
>    We agree that evaluations on additional datasets are valuable. In our supplementary, we currently show results on spleen data as well (as mentioned on L266-267), where our method still produces good results. To highlight this more, we have moved this section into the main paper under section 4.1.
>
>    Unfortunately, we were not able to provide evaluations on further datasets in the time frame of the rebuttal. However, we believe that our current results already shows that our method provides a well performing shape model that can be applied to a wide variety of tasks making it a useful contribution.

---

### Meta-Review · Area_Chair_tFPu · 2024-10-31

**Recommendation:** Accept (Poster)
**Confidence:** 4

**Metareview:**

The paper proposed a novel method for 3D shapes reconstruction by learning a mesh deformation. Reviewers found the proposed approach novel and conceptually interesting with solid experimental results and reproducibility.  They also commend great paper presentation and high quality of the manuscript.
Overall, this paper makes a significant contribution and is recommended for acceptance.

**Suggested Changes To The Recommendation:**

3: I agree that the recommendation could be moved up

---

### Decision · Program_Chairs · 2024-11-06

**Decision:**

Accept (Oral)

**Comment:**

Given the AC positive recommendation, we recommend an oral and a poster presentation given the AC and reviewers recommendations.